# Analysis of Influence of Floating-Deck Height on Oil-Vapor Migration and Emission of Internal Floating-Roof Tank Based on Numerical Simulation and Wind-Tunnel Experiment

**Gao Zhang [1], Fengyu Huang [1], Weiqiu Huang [1],\* , Zhongquan Zhu [2], Jie Fang [1], Hong Ji [1], Lipei Fu [1] and Xianhang Sun [1]**

1   Jiangsu Key Laboratory of Oil & Gas Storage and Transportation Technology, Changzhou University, Changzhou 213164, China; e9940610@gmail.com (G.Z.); hfy524320@163.com (F.H.); fangxyjoyce@sina.com (J.F.); jihong@cczu.edu.cn (H.J.); fulipeiupc@163.com (L.F.); sxh19871124@163.com (X.S.)

2   Yangzhong Environmental Sanitation Administration Agency, Yangzhong 212200, China; zzq0620@126.com

\*   Correspondence: hwq213@cczu.edu.cn

**Abstract:** Internal floating-roof tanks (IFRTs) are widely used to store light oil and chemical products. However, if the annular-rim gap around the floating deck becomes wider due to abrasion and aging of the sealing arrangement, the static breathing loss from the rim gap will be correspondingly aggravated. To investigate the oil-vapor migration and emissions from an IFRT, the effects of varying both the floating-deck height and wind speed on the oil-vapor diffusion were analyzed by performing numerical simulations and wind-tunnel experiments. The results demonstrate that the gas space volume and the wind speed of an IFRT greatly influence the vapor-loss rate of the IFRT. The larger the gas space volume, the weaker the airflow exchange between the inside and outside of the tank, thereby facilitating oil-vapor accumulation in the gas space of the tank. Furthermore, the loss rate of the IFRT is positively correlated with wind speed. Meanwhile, negative pressures and the vortexes formed on the leeward side of the tank. In addition, the higher concentration areas were mainly on the three vents on the downwind side of the IFRT. The results can provide important theoretical support for the design, management, and improvement of IFRTs.

**Keywords:** internal floating-roof tank; evaporation loss; diffusion; numerical simulation; wind tunnel experiment

## 1. Introduction

Internal floating-roof tanks (IFRTs) are widely used to store light oil, oil products, and chemical products. These tanks can significantly reduce the evaporating area of the stored liquids owing to the arrangement of a floating deck, thereby effectively restraining oil evaporation and reducing oil-vapor discharge from the tank. However, even if an IFRT is equipped with a sealing device on the annular-rim gap between the floating deck and the inner tank wall, the oil surface and the tank gas space cannot be completely isolated for the convenience of the floating deck moving up and down [1]. In addition, when the elasticity of the rim seal gradually decreases, and the gap of the rim seal is widened with long-term usage and abrasion, oil evaporation from the rim gap will gradually increase. Furthermore, oil evaporation from the stored liquid into the tank gas space and vapor emissions from the tank gas space into the atmosphere will cause oil loss, environmental pollution, and potential fire hazards [2].

Therefore, the interior mechanism of oil-vapor migration and emissions in IFRTs must be analyzed to ensure their safe operation.

Scientists have conducted many relevant research works, where the combination of numerical simulations and experimental measurements have been widely used as important tools for studying the oil-evaporation loss mechanism. Pasley et al. [3], and Zhao et al. [4], numerically simulated and experimentally measured the distribution of wind speeds and flow field around an external floating-roof tank and above the floating deck, and they observed that the airflow had two different flow characteristics each for the floating deck at lower positions and higher positions, respectively. Uematsuet et al. [5,6], investigated the wind-force distribution and the buckling behavior for open-topped oil-storage tanks. Wang et al. and Huang et al. [7–9], and Karbasian et al. [10], compared the effects of different oil-collection methods, oil-collection rates, and initial oil-vapor-mass fractions on the oil-vapor-diffusion law in the storage tank during the collection process of oil products. Hou et al. [11], analyzed the evolution of the flow field, temperature field, pressure field, oil-concentration field, and evaporation rate during the refueling process. In addition, many researchers have also studied other factors affecting oil-vapor emissions, such as vehicle and ship loading operations [12,13], the temperature-change characteristics of oil products in a storage tank [14–17], and the characteristics of oil vapors emitted from oil depots [18–20]. Subsequently, the researchers proposed a series of methods for assessing oil-vapor emissions [21,22]. However, these previously conducted studies have paid little attention to oil-vapor migration and emissions from an IFRT. The oil loss in a storage tank under normal operating conditions can be roughly divided into three processes: (1) heat and mass transfer between the liquid phase and the gas phase in the storage tank; (2) oil-vapor migration in the gas phase (i.e., in the gas space) of an IFRT; (3) oil-vapor emission and diffusion from the gas space into the atmosphere. In this study, the oil-evaporation rates were measured using self-made wind-tunnel experimental measurements. Subsequently, the species transfer model and the realizable k–$\varepsilon$ model in the ANSYS Fluent software were used to simulate the oil-vapor diffusion process in an IFRT, following which the effects of floating-deck heights and ambient wind speeds were investigated.

## 2. Methodology

### 2.1. Experimental Protocol

To measure the oil-evaporation rate and the wind speed in the tank under different operating conditions, a model IFRT (1000 m$^3$) was built according to the length ratio of 32:1, as depicted in Figure 1. The inner diameter, wall height, roof height, and rim gap of the tank were 360, 375, 39, and 6 mm, respectively. The size of the vents of the model tank was designed according to the selection principle of the prototype tank as follows:

$$B \geq 0.06D \tag{1}$$

where $B$ denotes the total effective ventilation area of the model-tank vents, m$^2$, and $D$ is the inner diameter of the tank, m. The effective ventilation area of the model-tank vents should be greater than 0.69 m$^2$, and the number of vents should not be less than four. Eventually, the model tank was equipped with four vents, each being 19 mm wide and 10 mm high. Further, the vents were evenly placed at the tank wall near the tank roof. In addition, four different floating-deck heights measuring 88, 176, 264, and 312 mm were set for the IFRT.

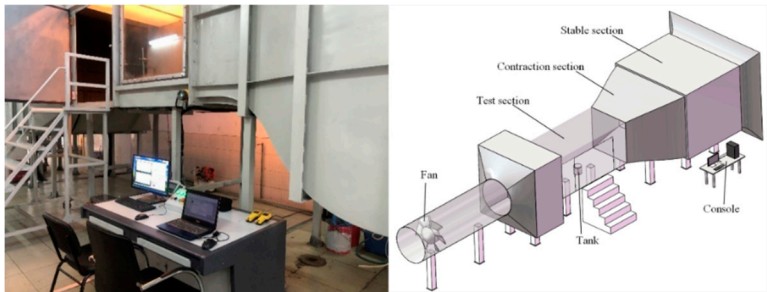

**Figure 1.** Wind tunnel for the experiment and simulation.

Due to the uncertainty of gasoline composition, the authors used n-hexane as the experimental oil for the convenience of research. The mass-difference method was used to measure the mass change of n-hexane in a certain period of time. The wind fields were generated using a self-made wind tunnel (DFWT-10), the size of the test section is 1.5 (H) × 1.5 (W) × 3 m (L), as depicted in Figure 1. The pitot tube anemometer is composed of a static pressure pitot tube (Kimo Instruments Co., Ltd., Bordeaux, France) and a digital micro-manometer (Yokogawa Electric Corporation, Musashino, Japan), which is used for real-time monitoring and feedback of wind speed in the wind tunnel [23]. The wind tunnel could provide wind fields with wind speeds ranging from 0.5 to 20 m·s$^{-1}$. The wind speed, temperature, and humidity were measured using a hot-wire anemometer (TES-1341, TES Co., Ltd., Taiwan, China, having the following specifications: wind-speed range of 0–30 m·s$^{-1}$ with a resolution of 0.01 m·s$^{-1}$; temperature range of −10 to −60 °C with a resolution of 0.01 °C; humidity range of 10–95% Relative humidity (RH) with a resolution of 0.1% RH). The evaporation loss in the IFRT could be automatically measured using a high-precision electronic balance (WT-30000-1B, Xinheng Electronics Co., Ltd., Shanghai, China, having a range of 0–30 kg with a resolution of 0.01 g). The mass of n-hexane was measured before and after the experiment, and the change in the mass could be calculated as the mass loss of n-hexane during this period. Subsequently, the variation in the mass per unit time can also be calculated as the mass-loss rate of n-hexane. The experiments were arranged under the following conditions: the ambient temperature of 13 °C, and wind speeds of 4.36 and 6.36 m·s$^{-1}$. The wind direction was directly opposite one of the four vents on the tank wall, and the mass-loss rates of n-hexane under different wind speeds were measured respectively.

*2.2. Theoretical Models for Oil-Vapor Diffusion*

2.2.1. Basic Governing Equations

The single-phase multicomponent diffusion problem without chemical reaction needs to be solved using the mass- and momentum-conservation equations. The mass-conservation equation can be written as follows:

$$\frac{\partial \rho}{\partial t} + \nabla \cdot \left( \rho \vec{v} \right) = S_m \tag{2}$$

where $S_m$ denotes the mass source term (in kg·m$^{-3}$·s$^{-1}$), and $\vec{v}$ the velocity vector (m·s$^{-1}$).

The momentum-conservation equation can be written as follows:

$$\frac{\partial}{\partial t}\left( \rho \vec{v} \right) + \nabla \cdot \left( \rho \vec{v} \vec{v} \right) = -\nabla p + \nabla \cdot \left[ \mu \left( \nabla \vec{v} + \nabla \vec{v}^T \right) \right] + \rho \vec{g} + \vec{F} \tag{3}$$

where $p$ denotes the static pressure (Pa), and $\rho \vec{g}$ and $\vec{F}$ denote the gravitational body force and external body force (both in N·m$^{-3}$), respectively. Term $\mu$ denotes the dynamic viscosity (Pa·s).

The energy-conservation equation can be written as follows:

$$\frac{\partial}{\partial t}(\rho E) + \nabla \left[ \vec{v}(\rho E + p) \right] = \nabla \cdot \left( k_{eff} \nabla T \right) + S_h \tag{4}$$

where energy $E$ (J·kg$^{-1}$) and temperature $T$ (K) are mass average variables, $k_{eff}$ the effective thermal conductivity (W·m$^{-1}$·K$^{-1}$), and $S_h$ the energy source term (J·m$^{-3}$·s$^{-1}$).

In turbulent flows, the species-conservation equation can be written as follows:

$$\frac{\partial}{\partial t}(\rho Y_i) + \nabla \cdot \left(\rho \vec{v} Y_i\right) = -\overline{\nabla} \cdot \vec{j}_i + S_i \tag{5}$$

$$\vec{J}_i = -\left(\rho D_{i,m} + \frac{\mu_t}{Sc_t}\right)\nabla Y_i - D_{T,i}\frac{\nabla T}{T} \tag{6}$$

where $Y_i$ denotes the local mass fraction of species $i$, and $\vec{J}_i$ denotes the diffusion flux of species $i$ (kg·m$^{-2}$·s$^{-1}$), which arises because of the gradients of concentration and temperature. Furthermore, $S_i$ denotes the source term (kg·m$^{-3}$·s$^{-1}$), $D_{i,m}$ the mass-diffusion coefficient for species $i$ in the mixture, $D_{T,i}$ the thermal-diffusion coefficient, and $Sc_t$ the turbulent Schmidt number.

### 2.2.2. Turbulence Equation

The oil vapors diffused from the IFRT are greatly affected by the ambient wind, and the oil-diffusion process is a complex unsteady turbulent flow. Therefore, the realizable $k$–$\varepsilon$ model with better turbulent accuracy is selected. The following is the modeled transport equation for $k$ and $\varepsilon$ in the realizable $k$–$\varepsilon$ model:

$$\frac{\partial}{\partial t}(\rho k) + \frac{\partial}{\partial x_j}(\rho k u_j) = \frac{\partial}{\partial x_j}\left[\left(\mu + \frac{\mu_t}{\sigma_k}\right)\frac{\partial k}{\partial x_j}\right] + G_k + G_b - \rho\varepsilon - Y_M + S_k \tag{7}$$

and

$$\frac{\partial}{\partial t}(\rho\varepsilon) + \frac{\partial}{\partial x_j}(\rho\varepsilon u_j) = \frac{\partial}{\partial x_j}\left[\left(\mu + \frac{\mu_t}{\sigma_\varepsilon}\right)\frac{\partial\varepsilon}{\partial x_j}\right] + \rho C_1 S_\varepsilon - \rho C_2\frac{\varepsilon^2}{k + \sqrt{v\varepsilon}} + C_{1\varepsilon}\frac{\varepsilon}{k}C_{3\varepsilon}G_b + S_\varepsilon \tag{8}$$

where

$$C_1 = max\left[0.43, \frac{\eta}{\eta + 5}\right], \ \eta = S\frac{k}{\varepsilon}, \ S = \sqrt{2S_{ij}S_{ij}}$$

In the above-mentioned three equations, $k$ denotes the turbulence kinetic energy(m$^{-2}$·s$^{-2}$) and $\varepsilon$ the dissipation rate (m$^2$·s$^{-3}$). Furthermore, $G_k$ and $G_b$ denote the generation of the turbulence kinetic energy due to the mean velocity gradients and buoyancy, respectively (kg·m$^{-1}$·s$^{-1}$). $Y_M$ denotes the contribution of the fluctuating dilatation in compressible turbulence to the overall dissipation rate (kg·m$^{-1}$·s$^{-1}$). $C_2$ and $C_{1\varepsilon}$ are constants. $\sigma_k$ and $\sigma_\varepsilon$ are the turbulent Prandtl numbers for $k$ and $\varepsilon$, respectively. $S_k$ denotes the source of the turbulence kinetic energy (kg·m$^{-1}$·s$^{-3}$) and $S_\varepsilon$ the source of the dissipation rate (kg·m$^{-1}$·s$^{-4}$).

### 2.3. Physical Model and Methodology

In wind-engineering calculations, the blocking ratio is usually employed as the basis for setting the cross-sectional area of the computational domain. The blocking ratio $r_b$ is defined as follows:

$$r_b = \frac{A_m}{A_C} \tag{9}$$

where $A_m$ and $A_c$ denote the maximum windward area of the model and the cross-sectional area of the computational domain (both in m$^2$), respectively. If the blocking ratio is less than 5%, the simulation results of the flow field may not be affected by each boundary [24–26]. An experimental model that is too small will increase the difficulty of experimental data measurement and lead to greater errors. We increased the size of the tank in the wind tunnel experiment to make the measurement results more exact. The blockage rate of the wind tunnel experimental model is 6%, calculated by Equation (9). To obtain the numerical-simulation results independent of the computational domain, the computational domain was selected as a cuboid (see Figure 2) of dimensions 2 (H) × 2 (W) × 6 (L),

and the size of the model tank (see Figure 3) was consistent with that of the experimental model. A three-dimensional grid-type model of the IFRT was established using ICEM CFD 14.5 software (ANSYS Inc., Pittsburgh, PA, USA).

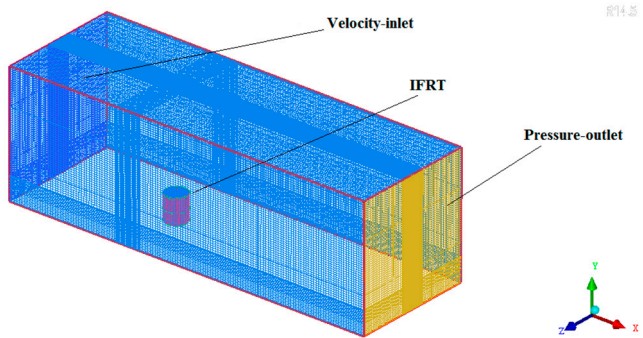

**Figure 2.** Grid model for oil-vapor diffusion of the Internal floating-roof tank (IFRT).

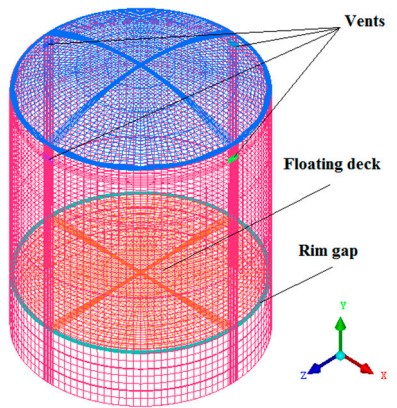

**Figure 3.** Grid model for the IFRT.

In the numerical calculation, the locations where the physical parameters in the flow field change require particular attention. These locations include the annular-rim gap between the floating deck and the inner tank wall of the IFRT, and the four vents on the tank wall. The grid of these regions must be partly encrypted to obtain more accurate numerical solutions. In this study, a structured grid type was designed for the computational domain. The number of cells was approximately 2 million; the quality of the grid was above 0.6, and its independency was also examined. The numerical-calculation procedure was performed using the commercial software package, ANSYS Fluent 14.5. Based on the species-transfer model, the measured oil-vapor-loss rates as the initial values of the mass transfer rate in the annular rim gap. The computational-domain inlet was set as the velocity boundary, and the computational-domain outlet was set as the pressure outlet. The vents on the tank wall were set as the interior boundary. The pressure and velocity were coupled using the SIMPLE scheme, and the spatial discretization of pressure was based on the Standard algorithm. The momentum-conservation equations were discretized using a second-order upwind scheme to reduce the numerical diffusion.

In wind tunnel experiments, we chose the inner floating roof tank as the experimental model. Re was calculated as 104,495 when the characteristic length is the model diameter, the wind velocity is 4.36 m·s$^{-1}$, the atmospheric density is 1.205 kg·m$^{-3}$, and the dynamic viscosity is $1.81 \times 10^{-5}$ Pa·s. Actually, the model is 1/32 of the actual size, which requires that the inflow speed of the wind tunnel must be 32 times the actual wind speed to achieve Re equality, it is obviously difficult to achieve. Thus, it is unrealistic to strictly achieve Re equivalence. Some researchers [27,28] believe that the speed distribution would be independent of Re if the studied model Re is over the critical Re, which is Re-independence. Based on this, a good agreement was achieved between the CFD simulation and wind-tunnel experiment.

## 3. Results and Analysis

### 3.1. Verification of the Flow Field

Before using the numerical model, the consistency of the flow field of the numerical simulation must be experimentally verified. This experiment studied the speed boundary layer in the wind tunnel when no tank was placed in the wind tunnel. When the flow speed of the wind tunnel is controlled at 4 m·s$^{-1}$, the wind speed in the X direction was measured at the center line of the section 500 mm away from the entrance of the wind tunnel test section. The measuring points are 20, 40, 60, 80, 100, 150, 200, 250, 300, 350, 400, 450, 500, 550, and 600 mm from the bottom, respectively. The experimental results are compared with the simulation results, listed in Figure 4. The boundary layer is 60 mm, so the tank is located in a uniform flow. According to Figure 4, the numerical simulation can better simulate the boundary layer conditions of the wind tunnel experiment.

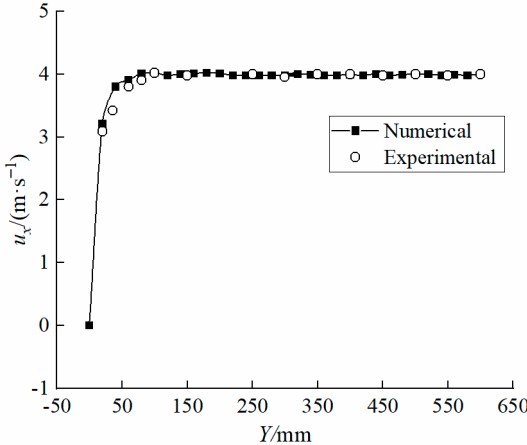

**Figure 4.** Comparison of wind speed in the X direction between the wind-tunnel experiment and numerical simulation on the inlet of 500 mm.

Four points on the tank circumference near the four vents were selected (assuming the center of the tank bottom as the origin). The ambient wind speed was set to 4.36 m·s$^{-1}$, and the floating-deck height was set to 176 mm. The hot-wire anemometer (TES-1341, TES Co., Ltd., Taiwan, China) was used to measure the wind speeds at these points. Each point was measured five times, and the wind-speed results are presented in Table 1.

**Table 1.** X-axis wind speeds at each measuring point of the tank.

| Measuring Point and Its Coordinates, mm | Measured Value, m·s$^{-1}$ | | | | | | Simulated Value, m·s$^{-1}$ | Error,% |
|---|---|---|---|---|---|---|---|---|
| | Test 1 | Test 2 | Test 3 | Test 4 | Test 5 | Average | | |
| Point 1 (190, 360, 0) | 0.54 | 0.53 | 0.65 | 0.72 | 0.61 | 0.61 | 0.48 | 21.3 |
| Point 2 (−190, 360, 0) | 2.63 | 2.71 | 2.62 | 2.64 | 2.68 | 2.66 | 2.57 | 3.4 |
| Point 3 (0, 360, 190) | 5.62 | 4.65 | 4.70 | 4.71 | 4.51 | 4.84 | 5.40 | 11.6 |
| Point 4 (0, 360, −190) | 4.76 | 4.64 | 4.62 | 4.51 | 4.66 | 4.64 | 5.39 | 16.7 |

It can be seen from Table 1 that the average wind speeds at Points 3 and 4 are approximately equal, both of which are slightly greater than the ambient wind speed of 4.36 m·s$^{-1}$ owing to the effect of the tank body. However, because the airflow was blocked by the tank wall, the average wind speed at Point 2 (which lies in the windward side of the tank) is lower than the ambient wind speed. In addition, the average wind speed at Point 1 (which lies in the back side of the tank) is the smallest. The four measured wind-speed values at Points 1 to 4 basically accord with the distribution law of the flow field around the tank. However, the measured values and the numerically simulated values are slightly different from each other. The reasons for the difference are as follows: (1) although the

flow field is stabilized in general, the uncertainty of turbulence exists partly; the physical parameter (i.e., wind speed) in the flow field fluctuates within a certain range; the measured values are also slightly different at different moments; (2) the error in the measuring instrument itself and the interference of the measuring instrument with the flow field can also cause measurement errors. Therefore, the consistency of the flow field of the numerical simulation was verified using the experiment, and the numerical-simulation model can be considered to be relatively suitable for oil-vapor-diffusion simulation of an oil tank.

### 3.2. Influence of Different Floating-Deck Heights on N-Hexane-Loss Rates

It can be seen from the experimental results in Table 2 that the loss rates of n-hexane can be influenced greatly by varying both floating-deck heights and wind speeds. In general, the floating-deck heights of the IFRT were positively correlated with the evaporation rates of n-hexane. However, the loss rates increased as the floating-deck heights and wind speeds increased. For the ambient wind speeds of 4.36 and 6.36 m·s$^{-1}$, setting the floating-deck height to 312 mm (the highest height), the loss rates of n-hexane reached $4.139 \times 10^{-6}$ and $5.960 \times 10^{-6}$ kg·s$^{-1}$, respectively. However, upon setting the floating-deck height to 88 mm (the lowest height), the loss rates were only $2.189 \times 10^{-6}$ and $3.773 \times 10^{-6}$ kg·s$^{-1}$, respectively.

**Table 2.** Loss rates of n-hexane measured in model tanks *.

| Floating-Deck Height, mm | Full Coefficient | r1/($10^{-6}$ kg·s$^{-1}$) | r2/($10^{-6}$ kg·s$^{-1}$) |
|---|---|---|---|
| 88 | 0.24 | 2.189 | 3.773 |
| 176 | 0.48 | 2.719 | 4.453 |
| 264 | 0.71 | 3.560 | 5.272 |
| 312 | 0.84 | 4.139 | 5.960 |

*: r1, r2—The loss rates measured for the wind speed of 4.36 and 6.36 m·s$^{-1}$, respectively.

The reasons for this phenomenon are as follows: the evaporation of n-hexane is driven by a diffusion process that is induced by the gradient of n-hexane-vapor concentration at the liquid surface. The larger the gas space inside the tank, the slower the gas flow, and thus the less frequent the gas exchange between the tank and the atmosphere. Therefore, the n-hexane-vapor concentration remains at a relatively stable level, and the driving force for n-hexane evaporation was reduced. Conversely, the smaller the gas space inside the tank, the more frequent the gas exchange between the tank and the atmosphere. As a result, the n-hexane vapor in the tank released into the atmosphere, resulting in fresh air entering the tank from the vents. The n-hexane-vapor concentration in the tank was maintained at a low value, increasing the n-hexane-vapor concentration difference between the gas space and the gas–liquid interface, thereby aggravating the evaporation of n-hexane.

### 3.3. Simulation of the Large IFRT

To investigate the concentration-distribution characteristics of the light oil products in the actual IFRT, the authors continued to simulate the oil-vapor-diffusion process in a large IFRT in the length ratio of 1:1 for a 1000 m$^3$ IFRT. N-hexane was still chosen as the stored oil product. The inner diameter, wall height, roof height, and rim gap of the tank were 11.5, 12.0, 1.254, and 0.2 m, respectively. The outlet boundary of the flow field was set as the pressure outlet boundary condition while the gap between the floating deck and the tank wall was set as the mass-flow boundary condition. The tank bottom, tank wall, and floating deck were all set as no-slip boundaries and the ambient temperature was set at 13 °C. The mass fraction of the saturated concentration of n-hexane when it diffuses in the gap of the floating disk is 0.30, according to the calculation of the saturated vapor pressure of n-hexane at 13 °C. We took the k–ε turbulence model to describe the diffusion process of the oil vapor.

### 3.3.1. Flow-Field Simulation

The streamtraces inside the tank with different floating-deck heights for the ambient wind speed of 4.36 m·s$^{-1}$ were simulated, as depicted in Figures 5–8. As can be seen from Figure 5, in the larger gas space of the tank, a clockwise vortex was formed, and the streamtraces of the vortex resembled the "0" type. As the gas space became smaller, the streamtraces of the vortex in the tank gradually assumed the shape of a flat ellipse (see Figures 6 and 7). Finally, the gas space became sufficiently small, following which the vortex mentioned above was divided into several small vortexes; it means that the airflow in the gas space was faster and more disorderly at this moment. Furthermore, the streamtraces outside the tank were simulated, as depicted in Figure 9. It can be seen from the figure that a part of the airflow from the inlet entered the tank and that the other part of the airflow bypassed the tank. Moreover, the airflow behind the tank did not flow in the original direction, and many vortexes were generated there.

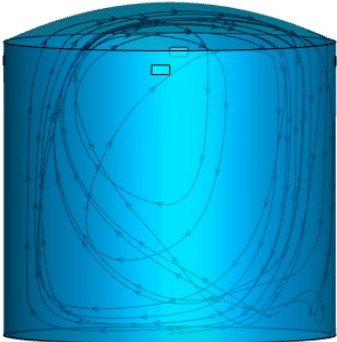

**Figure 5.** Streamtraces inside the tank for the floating-deck height of 2.82 m.

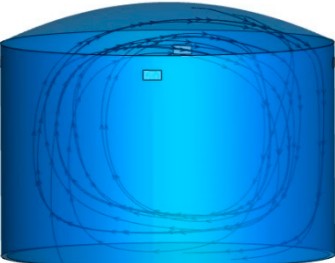

**Figure 6.** Streamtraces inside the tank for the floating-deck height of 5.63 m.

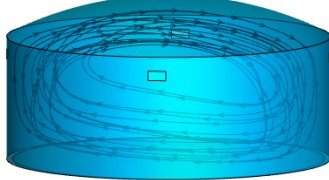

**Figure 7.** Streamtraces inside the tank for the floating-deck height of 8.45 m.

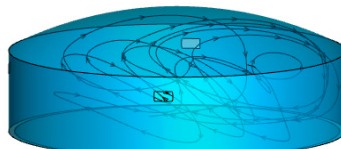

**Figure 8.** Streamtraces inside the tank for the floating-deck height of 9.98 m.

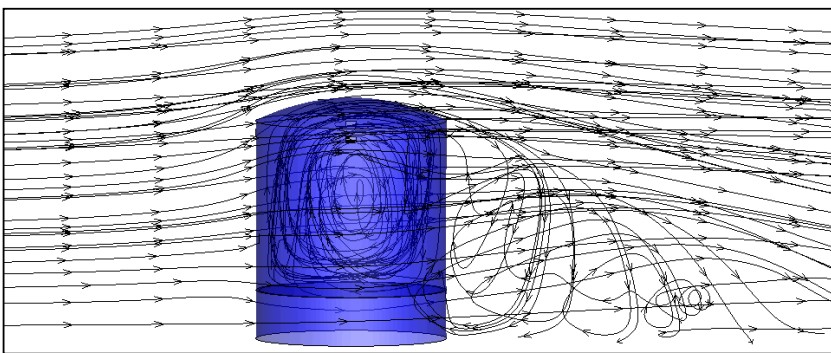

**Figure 9.** Streamtraces inside and outside the tank.

The diffusion of the n-hexane vapors evaporating from the tank was greatly affected by the ambient wind. In the numerical simulation, the ambient wind was set along the X-axis, and the wind speeds in the inlet were set to 4.36 and 6.36 m·s$^{-1}$, respectively. The velocity-contour distributions both inside and outside the IFRT in the XY plane are depicted in Figures 10 and 11. The results demonstrate that the airflow was obstructed by the tank at the windward side and that the velocity of the airflow was gradually reduced from 4.36 to 0 m·s$^{-1}$. Combined with the pressure-cloud diagram at the windward side of the tank wall in Figure 12, it can be concluded that the pressure on the windward side of the tank body increased because the wind speed translated into stagnation pressure. On the surface of the tank roof, the flow cross-section of the ambient wind shrunk, causing a sudden increase in the wind speed and negative pressure (see Figure 13). The wind speed near the tank roof exceeded the ambient wind speed, and the maximum wind-speed value reached 5.5 and 8.0 m·s$^{-1}$, respectively. On the leeward side of the tank wall, the wind-speed iso-surface continuously shrunk toward the bottom of the tank because of the obstruction of the tank body (see Figures 10 and 11). The minimum value of the wind speed was approximately 0 m·s$^{-1}$ at the bottom of the leeward side of the tank wall. Compared to the windward side, the region of lower wind speed was larger in the leeward side. Combined with the wind-flow diagram of the inside and outside of the tank (see Figure 9) and the tank-wall pressure-cloud diagram on the leeward side of the tank (see Figure 13), it can be seen that negative pressure and vortexes appeared near the leeward side of the tank.

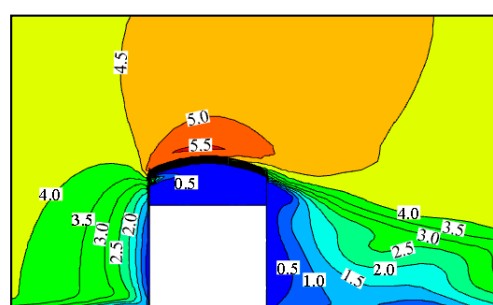

**Figure 10.** Velocity distribution of the IFRT under the ambient wind speed of 4.36 m·s$^{-1}$.

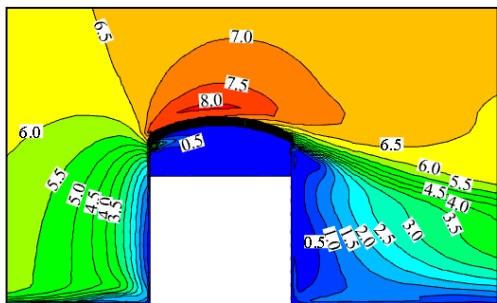

**Figure 11.** Velocity distribution of the IFRT under the ambient wind speed of 6.36 m·s$^{-1}$.

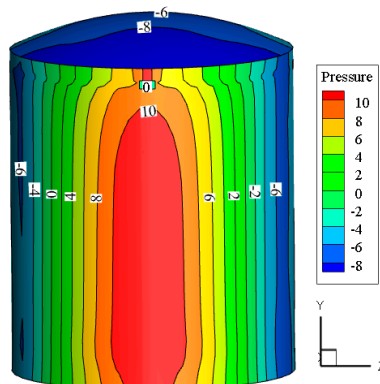

**Figure 12.** Pressure-cloud diagram for the windward side of the tank wall (Pa).

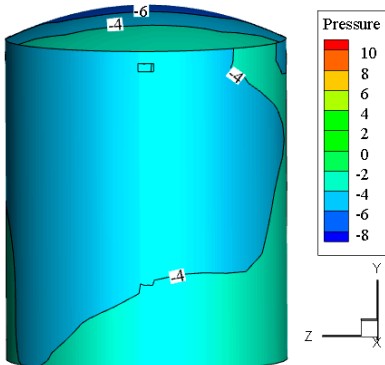

**Figure 13.** Pressure-cloud diagram for the leeward side of the tank wall (Pa).

Several monitored points were set along the central axis of the tank to obtain the data of wind speed inside the tank, as depicted in Figure 14. The wind speed in the tank was low because the sizes of the vents were too small relative to the size of the gas space. It can be seen from Figure 15 that for the ambient wind speed of 4.36 m·s$^{-1}$, the velocities of the airflow in the gas space of the tank were less than 0.6 m·s$^{-1}$. In addition, upon lowering the floating-deck height, the speeds measured from the monitored points changed greatly. The wind speeds at the points near the roof and floating deck were higher than those at other monitored points, while the wind speed at the center of the gas space was low. The minimum wind speed in the gas space was approximately 0.05 m·s$^{-1}$. Upon increasing the height of the floating deck, the low-velocity region gradually reduced. When the floating-deck height reached a certain height, the velocity distribution became irregular. This irregularity reconfirmed the conclusions drawn previously that upon changing the floating-deck height, the streamtraces of the airflow in the tank also changed.

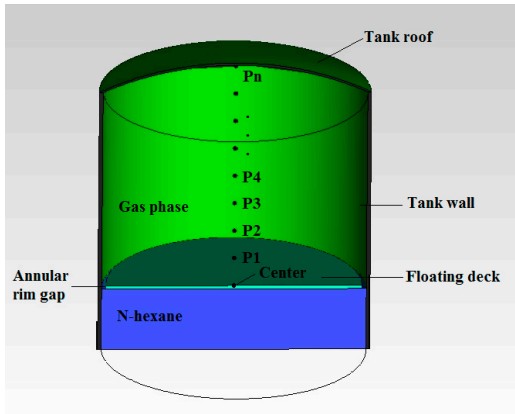

**Figure 14.** Location of monitored points.

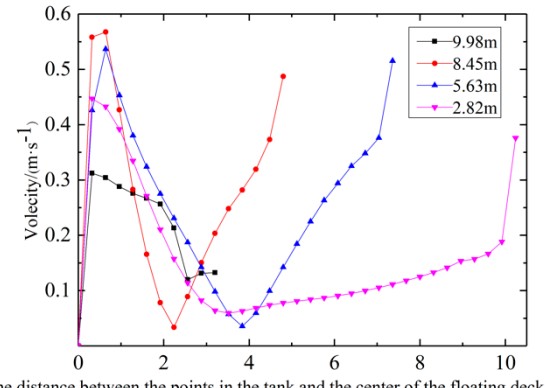

**Figure 15.** Velocities inside the tank for different floating-deck heights.

### 3.3.2. Concentration Distribution of N-Hexane Vapor in the Tank

The concentration distribution at different floating-deck heights in the tank is depicted in Figure 16. It can be seen that upon increasing the wind speed under a fixed floating-deck height, the concentration of n-hexane in the gas space in the tank becomes low. The higher concentration of n-hexane vapors was mainly distributed near the rim gap, and the concentration distribution of n-hexane vapors in the upper gas space of the tank was relatively uniform. Upon setting the floating-deck height to 2.82 m, because of the resulting large gas space in the tank, the airflow inside and outside the tank is exchanged infrequently, and thus the concentration of n-hexane in the tank becomes evenly distributed. Upon increasing the floating-deck height to 5.63 m, the n-hexane-vapor concentration was affected by the air inflow, which formed a large vortex in the tank. Upon further raising the floating-deck height to 8.45 m, the gas exchange between the inside and outside of the tank was accelerated, and thus the large vortex in the tank was destroyed (wind speed = 6.36 m·s$^{-1}$) or compressed (wind speed = 4.36 m·s$^{-1}$) because of the reduction of the gas space in the tank. Furthermore, upon increasing the floating-deck height to 9.98 m, the n-hexane vapors were mainly concentrated in the area near the gap. In addition, n-hexane vapors at other regions were quickly discharged out of the tank along with the airflow, and it became difficult for the n-hexane vapors to stay in the tank.

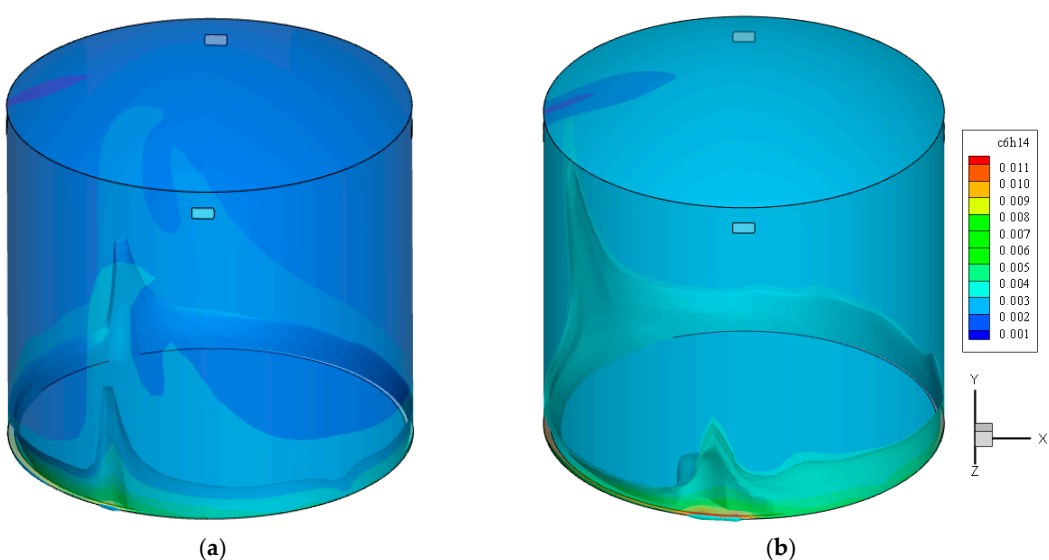

(**a**)  (**b**)

**Figure 16.** *Cont.*

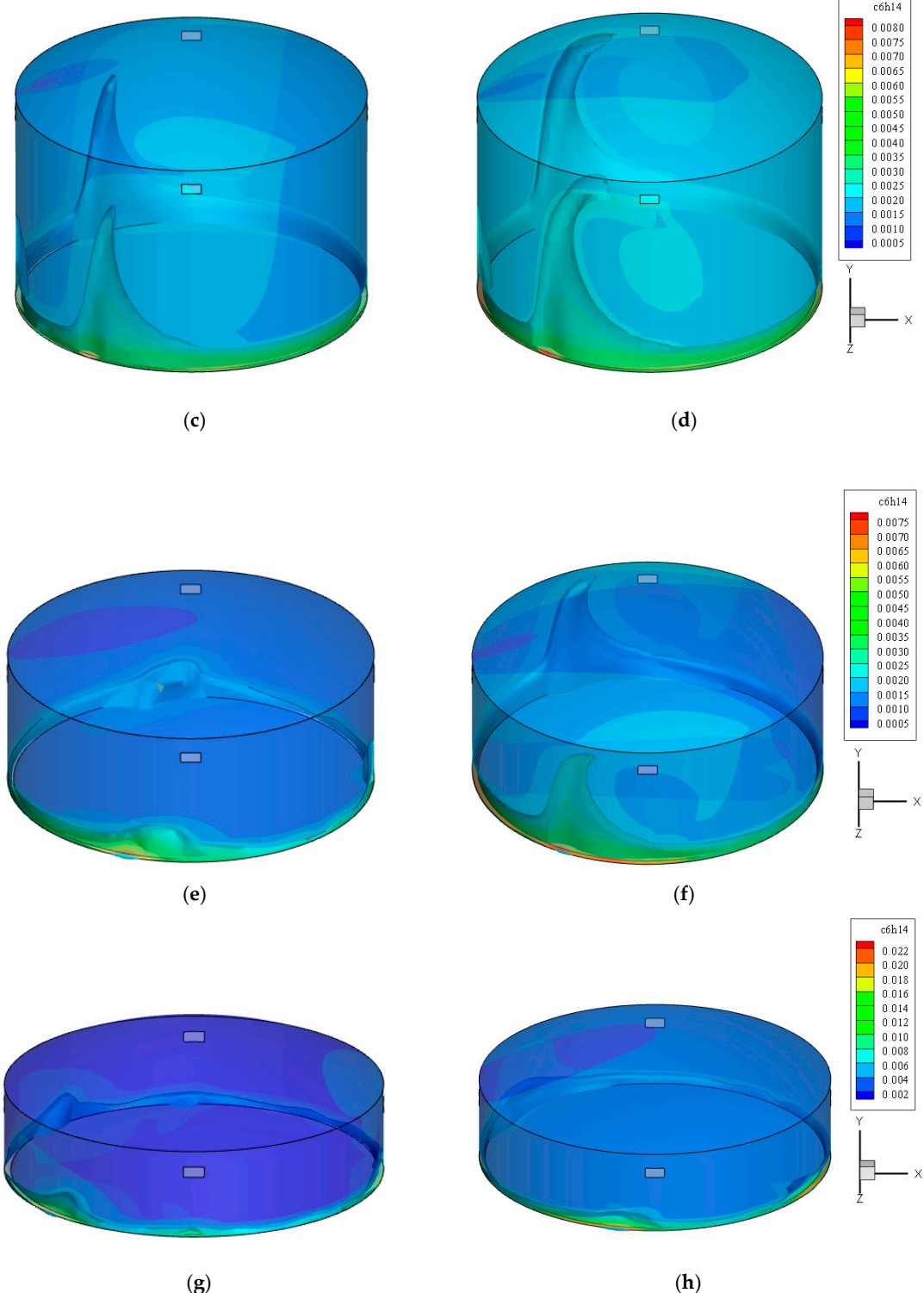

**Figure 16.** Distribution of n-hexane vapors (mass fraction) in the IFRT at different floating-deck heights. (**a**) Wind speed = 4.36 m·s$^{-1}$, floating-deck height = 2.82 m. (**b**) Wind speed = 6.36 m·s$^{-1}$, floating-deck height = 2.82 m. (**c**) Wind speed = 6.36 m·s$^{-1}$, floating-deck height = 5.63 m. (**d**) Wind speed = 4.36 m·s$^{-1}$, floating-deck height = 5.63 m. (**e**) Wind speed = 6.36 m·s$^{-1}$, floating-deck height = 8.45 m. (**f**) Wind speed = 4.36 m·s$^{-1}$, floating-deck height = 8.45 m. (**g**) Wind speed = 6.36 m·s$^{-1}$, floating-deck height = 9.98 m. (**h**) Wind speed = 4.36 m·s$^{-1}$, floating-deck height = 9.98 m.

The n-hexane-vapor emission from the IFRT was also investigated; the floating-deck height was at 9.98 m, wind speed 4.36 m·s$^{-1}$, and the concentration distribution as depicted in Figure 17. It can be seen that the n-hexane vapors discharged from the vents on the front and rear sides of the tank were diffused along the wind direction and that the vapors discharged from the leeward-side vent spread around the vent. Irrespective of the front and rear sides or the leeward side of the tank (the vent on the windward side is the airflow inlet), the mass fraction of n-hexane vapors near the vents reached 0.002, and such high concentration of vapors discharged into the atmosphere do not meet environmental safety requirements and will cause air pollution. Therefore, if equipping the vents with an oil-vapor-recovery unit is necessary, priority should be given to the above-mentioned three vents.

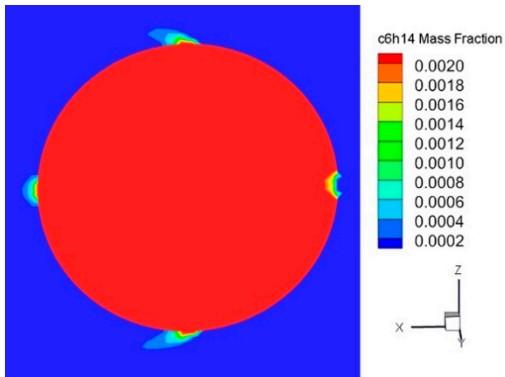

**Figure 17.** Distribution of n-hexane-vapor concentration (mass fraction) near the vents.

## 4. Conclusions

The evaporative loss rates of n-hexane in the IFRT model and the wind speeds of gas space above the floating deck inside the tank were measured using wind-tunnel experiments and also simulated using ANSYS Fluent software. The conclusions drawn from the research results can be summarized as follows:

(1) Based on numerical simulation and the wind-tunnel experiments, the oil-vapor diffusion process in the IFRT was simulated and was then verified to be relatively suitable to the oil-vapor-diffusion simulation for different sizes of IFRTs. This further revealed the law of mass transfer between the oil vapors and air for the evaporation and diffusion process in the IFRTs.

(2) Different floating-deck heights of the IFRT corresponded to different loss rates of n-hexane. The larger the gas space inside the tank, the weaker the airflow exchange between the inside and outside of the tank became. Therefore, the gradient of the n-hexane-vapor in the tank was lower, thereby reducing the driving force for n-hexane evaporation.

(3) Outside the tank, the direction of the ambient wind would change suddenly while bypassing the tank. In addition, negative pressure and vortexes were generated on the leeward side of the tank. Inside the tank, the n-hexane-vapor distribution was relatively uniform at lower floating-deck heights. Upon increasing the floating-deck height, a large vortex formed in the tank, intensifying the airflow disturbance in the tank. Upon further increasing the floating-deck height, the aforementioned large vortex was divided into several small vortexes, and the airflow in the gas space became faster and more disorderly than that in the previous floating-height condition.

(4) The higher vapor-concentration regions near the vents, compared with other regions in the tank, are mainly concentrated on the front and rear sides and the leeward side of the IFRT. Therefore, if it is decided to employ an oil-vapor-recovery unit, more attention should be paid to the regions having higher vapor concentrations.

**Author Contributions:** Conceptualization, G.Z., W.H.; software, F.H.; formal analysis, W.H., G.Z., F.H., L.F. and J.F.; investigation, Z.Z.; data curation, W.H., G.Z., and H.J.; writing—original draft preparation, G.Z., W.H., and X.S.; writing—review and editing, G.Z., W.H., F.H., L.F. and Z.Z.; supervision, W.H.; funding acquisition, W.H. All authors have read and agreed to the published version of the manuscript.

**Funding:** This research was funded by the National Natural Science Foundation of China (No. 51574044 and No. 51804045), the Key Research and Development Program of Jiangsu Province (Industry Foresight and Common Key Technology) (No. BE2018065), the Sci & Tech Program of Changzhou (No. CJ20180053) and Postgraduate Research & Practice Innovation Program of Jiangsu Province (NO. SJCX19_0668).

**Conflicts of Interest:** The authors declare no conflict of interest.

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
