# Peer review of "Analysis of Influence of Floating-Deck Height on Oil-Vapor Migration and Emission of Internal Floating-Roof Tank Based on Numerical Simulation and Wind-Tunnel Experiment"

_processes, doi:10.3390/pr8091026_

Round 1

Reviewer 1 Report

Very strange, there is no comparison between wind tunnel experiment and CFD simulations.

Moreover, I cannot find any meaningful graph explaining the results of physical parameters.

In this aspect, this paper is not ready for a JCR-class article yet.

Author Response

Dear Reviewer,

Reviewer 2 Report

Review of the article «Analysis of influence of floating-deck height on oil-vapor migration and emission of internal floating-roof tank based on numerical simulation and wind-tunnel experiment» by Gao Zhang, Fengyu Huang, Weiqiu Huang, Zhongquan Zhu, Jie Fang, Hong Ji, Lipei Fu, Xianhang Sun.
This research is relevant and will be interest the wide class of readers. The presented article contains a lot of data and it is well written.

However, despite the advantages of this scientific research, I have the following remarks:

1. I recommend making drawings with curves more readable for readers.
The curves are too close together and it will be difficult to understand the data in the figure for the reader. On the axes of the drawings, the font size should be increased;

2. I found few typos in the text. All typos should be corrected by Authors;

3. I advise to reduce the article title. The presented title is too bulky.

For example "The complex processes study of oil vapor migration in a wind tunnel".

I recommend to the Editor-in-Chief to accept this article for publication after the minor revision.

Author Response

Dear Reviewer,

Reviewer 3 Report

The manuscript presents a concise sensitivity analysis using numerical simulations and wind-tunnel experiments. I believe that the manuscript can be accepted on its present form.

Author Response

Dear Reviewer,

Reviewer 4 Report

Summary: The paper describe numerical and experimental approaches to model n-heptane vaporization from a floating deck tank. This problem has many applications. Therefore, this work has a good practical value. Also, combining numerical and experimental analysis gives the readers a better confidence in the presented results. The authors has presented a good summary of the previous work and the details about the governing equations. However, the details about the numerical approach is lacking a little bit (see the comments below). I recommend to publish the paper after addressing the following comments.
Comments:
1) Include an actual picture (or an image of the CAD geometry) of the experimental tank as the second figure.
2) Lines 81-82 read “The wind fields were generated using a self-made wind tunnel”. Calibration of the wind tunnel is an important step in ensuring accurate measurements. Include any references that you have on the fabrication and calibration of the wind tunnel.
3) What is the cross-sectional area of the wind tunnel? Does this match with the dimensions of the computational domain given in lines 134-136? What is the percentage blockage of the model in the wind tunnel? Please include these in the revised manuscript.
4) What is the “quality” measure used in “…the quality of the grid was above 0.6” in line 144?
5) There is a big variation in measured and simulated results given in Table 1. This geometry is a blunt body, with a lot of unsteadiness and flow separation. To model this appropriately, one has to simulated this in an unsteady mode, and get the average values and the variation over time. Same with the experimental measurement too. Rerun the simulation using the transient model and include the data in the revised manuscript.
6) Concentration distribution of n-hexane vapor in the tank is given in Section 3.3.2. The results look interesting. However, the details about the model setup is missing. Please include the information such as the gap between the interior wall and the floating deck, how the concentration at the gap region is specified, how did the evaporation based on the flow properties and temperature is modeled, etc.

Author Response

Dear Reviewer,

Round 2

Reviewer 1 Report

  1. The details of a model setup in the wind tunnel should be described. It is not obvious whether the tank is located in a boundary layer or in a uniform flow. Either case, not only wind speed profile at the inlet but also turbulence profiles should be described. Because that the turbulent diffusion, which is a key mixing mechanism in the tank, is likely to depend on the turbulence level of the flow.
  2. There is any remark or consideration about the similarity of the scaled-down wind tunnel model and CFD simulation. Eventually, this results have to be extrapolated to a real scale but none of such consideration is found in this manuscript. Therefore, Table 1 and Figure 14 should be made as an non-dimensional form. I cannot believe that Reynolds number is not mentioned in a flow modeling and experiment paper.
  3. The most important aspect is, the academic novelty of this paper is not seen at all. They employed the commercial CFD software, and just made a comparison between experiment and CFD, but is not proper because they did not consider geometric and kinematic similarity between the wind tunnel model and CFD and the real object. 

Author Response

Dear Reviewer,
